# Psychological Wellbeing and Academic Experience of University Students in Australia during COVID-19

**DOI:** 10.3390/ijerph18030866

**Published:** 2021-01-20

**Authors:** Rachael H. Dodd, Kevin Dadaczynski, Orkan Okan, Kirsten J. McCaffery, Kristen Pickles

**Affiliations:** 1Faculty of Medicine and Health, School of Public Health, The University of Sydney, Sydney 2006, Australia; kirsten.mccaffery@sydney.edu.au (K.J.M.); kristen.pickles@sydney.edu.au (K.P.); 2Department of Nursing and Health Science, Fulda University of Applied Sciences, 36037 Fulda, Germany; kevin.dadaczynski@pg.hs-fulda.de; 3Centre for Applied Health Science, Leuphana University Lueneburg, 21335 Lueneburg, Germany; 4Interdisciplinary Centre for Health Literacy Research, Bielefeld University, 33615 Bielefeld, Germany; orkan.okan@uni-bielefeld.de

**Keywords:** COVID-19, wellbeing, students, university, education

## Abstract

COVID-19 has created significant challenges for higher education institutions and major disruptions in teaching and learning. To explore the psychological wellbeing of domestic and international university students during the COVID-19 pandemic, an online cross-sectional survey recruited 787 university students (18+ years) currently studying at an Australian university. In total, 86.8% reported that COVID-19 had significantly impacted their studies. Overall, 34.7% of students reported a sufficient level of wellbeing, while 33.8% showed low wellbeing and 31.5% very low wellbeing. Wellbeing was significantly higher in postgraduate students compared with undergraduate students. Future anxiety was significantly greater among undergraduate than postgraduate students. Multivariable regression models showed female gender, low subjective social status, negative overall learning experience or reporting COVID-19 having a huge impact on study, were associated with lower wellbeing in the first few months (May–July) of the pandemic. Supporting the health, wellbeing, and learning experiences of all students should be of high priority now and post-pandemic. Strategies specifically targeting female students, and those with low self-reported social status are urgently needed to avoid exacerbating existing disparities.

## 1. Introduction

The 2019–2020 coronavirus pandemic has prompted extraordinary measures to be implemented globally in an effort to reduce transmission of the virus [1]. In Australia, the number of new cases increased rapidly in late March, averaging 350 cases per day [2]. At this time, restrictions were placed on all international travel, state borders closed, returning international travellers entered mandated hotel quarantine, and social distancing rules were introduced. In July, the state of Victoria enforced mask wearing in public spaces following a “second wave” of COVID-19 infections [3,4]. To reduce the spread of the virus among younger and adult populations, Australian universities were closed nationally at the start of the academic year in March 2020, with on-campus learning suspended, educational and social events postponed or cancelled, and student accommodation facilities closed. As of 15 January 2021, Australia has reported 28,658 cases, 275 active cases, and 909 deaths [2]. There is currently very low incidence of COVID-19 across the country, with the majority of recent cases related to travellers in hotel quarantine.

At least initially, the pandemic raised significant challenges for higher education institutions and major disruptions in teaching and learning. Due to travel restrictions in place, many international and interstate students were unable to begin semester one (March 2020) as planned, with university courses and programs rapidly transitioning from face-to-face to online delivery to ensure the continuity of teaching and assessment. Online learning presents its own challenges, with many students experiencing lost learning opportunities if their chosen field of study was not amenable to exclusive online teaching, such as clinical work in medicine and health sciences [5]. There has been widespread media coverage on how the pandemic might impact graduation and career prospects in a global recession [6,7].

University students have been identified as a “very high risk population” for mental health difficulties [8]. Large studies conducted in Australian universities have reported elevated levels of generalized psychological distress and severe depressive symptoms when compared to population samples [9,10]. Studies in the United States and United Kingdom have reported similar findings among tertiary students [11,12]. Psychological distress negatively impacts student learning, participation, and their experience of university life, so it is important for universities to understand the student experience of particular stressors to better support their psychological wellbeing [13].

Understandably, heightened stress has been experienced by university students during COVID-19, with research reporting increased financial and psychological stress in doctoral candidates and medical students in Australia, compared to comparative pre-COVID-19 data [14,15]. Internationally, in a sample of over 7000 undergraduate students at a Chinese medical college, 25% experienced some level of anxiety due to COVID-19, including worry about economic stressors, academic delays, and effects of COVID-19 on daily life [16]. International students, who account for approximately 30% of higher education enrolments in Australia [17], may be particularly vulnerable to the impacts of COVID-19 due to their significant financial sacrifice and work visa restrictions [18] and not being eligible for government subsidies [19]. International students have also reported facing discrimination and isolation in some countries due to being deemed as potential coronavirus carriers [20], increasing risks to their mental health.

This research is part of an international collaboration investigating health literacy, health information seeking, future life perspectives, and mental health outcomes among university students [21]. The objective of this study was to explore the impact of COVID-19 on domestic and international university students in Australia—particularly their psychological wellbeing and learning experience—to identify those most in need of support. Our research questions were:How have university students in Australia been impacted by COVID-19? What were the characteristics of those affected?What sociodemographic factors were associated with low psychological wellbeing in university students during COVID-19?Was the impact of COVID-19 on university students’ learning associated with low wellbeing?

## 2. Materials and Methods

### 2.1. Participants: Selection and Exclusion Criteria

Participants were university students aged 18 years and over, able to read and understand English and currently studying at a university in Australia. Participants were excluded if they were not a university student, under 18 years of age and could not understand English.

### 2.2. Instruments

Sociodemographic variables, including age and gender (male, female, diverse) were collected. Students were asked questions about their university location, type of degree (Bachelor, Master, other), and student status (local, international, part or full time). Subjective social status (SSS) was assessed using the German version of the MacArthur Scale, which depicts a ladder with 10 steps [22]. Respondents were asked to position themselves at the step that best reflected their status on the social hierarchy with higher values indicating a higher social status.

Psychological wellbeing (World Health Organisation (WHO) wellbeing scale) [23], sense of coherence (SoC) (viewing one’s life as comprehensible, manageable and meaningful) [24] and future anxiety (state of uncertainty, worry and concern of unfavourable changes in a more remote personal future) [25] were measured. A detailed description of the instruments used and how they are scored can be found in Table 1.

Additional survey items measured the impact of COVID-19 on student’s living arrangements (Has your current location and/or living arrangements been impacted by COVID-19? No; Yes, e.g., I returned to my home country); study—particularly the move to online learning (nine statements; seven-point response scale: strongly agree–strongly disagree, e.g., “I find it more difficult to learn online than face-to-face”); employment (“Has your employment been impacted by COVID-19?” If yes, what was the change in your employment status due to COVID-19? 6 response options: e.g., lost job, reduced hours); and whether they had accessed any support.

### 2.3. Procedure

The online cross-sectional survey was carried out as part of a large-scale international survey led by Germany (the COVID-HL Consortium) in March (https://covid-hl.eu/). Worldwide, 49 countries and 94 researchers are involved in the Consortium and its survey among university students. The adapted Australian version of the survey was conducted between 29 May–6 July 2020. Participants were recruited via social media in Australia (Facebook, Instagram, Twitter) and university mailing lists to complete an online survey hosted by Qualtrics. This study was approved by The University of Sydney Human Ethics Committee (2020/343).

### 2.4. Data Analysis

Quantitative data were analysed using SPSS version 26. Descriptive analyses were conducted to give sample characteristics and independent t-tests were used to compare psychological variables by degree type and international versus domestic students and address research question 1. Chi squared tests were conducted for categorical variables (χ^2^). For all analyses, *p*-values < 0.05 were considered statistically significant. Multiple binary logistic regression models were performed for predictive factors of low wellbeing.

In the sociodemographic models, to address research question 2, sense of coherence and future anxiety were dichotomised as low versus high based on median split, and wellbeing combined indications for very low and low wellbeing versus high wellbeing as suggested by Topp et al. [26]. Based on previous research [27], subjective social status was categorised as low (1–4), medium (5–7), and high (8–10). Age was categorised as ≤20 years, 21–23 years, 24–26 years and ≥27 years. Model 1 included the sociodemographic variables of age, gender, subjective social status, language spoken at home and degree type. Model 2 included the sociodemographic variables with the sense of coherence scale, with model 3 additionally including future anxiety.

In the learning experiences models to address research question 3, the impact on study was collapsed into five groups. Model 1 included univariate analyses of impact on living, accessing support (psychological and living costs), overall learning experience and impact on study. Model 2 included overall learning experience and impact on study. Free text responses, specifying how living arrangements and study were impacted by COVID-19, and types of support received, were analysed thematically.

## 3. Results

### 3.1. Participants

Of 1326 students who agreed to participate, 137 were not eligible as they were not a university student, 399 did not complete the survey and 3 completed the survey in less than 4.6 min (<1/3 of the median of 13.7 min) and were therefore deemed ineligible. Therefore, a total of 787 participants were included in the analyses.

Table 2 shows the sample characteristics. The mean age was 25.7 years (SD 9.24; range 18–89 years) with majority female (67.0%). Most participants were born in Australia (65.3%) with 79.8% speaking English as their main language at home. Most participants were studying in New South Wales (57.1%), in the medicine and health sciences field (41%). In total, 62.5% were undergraduate, graduate certificate or diploma students, with 37.5% postgraduate students. Almost 23% were international students. Most of the sample reported their subjective social status as either medium (56.9%) or high (22.9%).

### 3.2. Impact of COVID-19 on Living and Study

Overall, students reported that COVID-19 had a huge impact on their studies in the last two weeks (86.8%), with this impact being mainly negative on their overall learning experience (70.9%; Table 3). The most common negative study-related impacts were finding it hard to interact with other students (84.6%) and teachers (74.6%) online, and it being more difficult to learn online than face-to-face (74.7%). However, almost half (47.6%) reported finding online learning less time consuming than face-to-face learning. In total, 55.3% indicated that their home environment supported online learning.

Younger students (≤23 years old), those with lower subjective social status, and speaking a language other than English at home, were more likely to report COVID-19 having a significant impact on their studies in the last 2 weeks compared to students aged over 24 years (X^2^ = 47.7, *p* < 0.001), with higher subjective social status (X^2^ = 27.6, *p* = 0.006), and English as their primary language (X^2^ = 14.3, *p* = 0.026). Males’ overall learning experience was more affected than females (X^2^ = 15.5, *p* < 0.001) as were those with lower compared to higher subjective social status (X^2^ = 14.4, *p* < 0.001). There was no significant difference between international and domestic students for either impact of COVID-19 on their studies or overall learning experience.

16.1% of students had accessed support offered by their university—such as assistance with postponing studies, wellbeing and counselling services, and/or financial support—with a greater proportion of students enrolled in a graduate certificate or diploma (26.1%) accessing support compared with Masters (22.9%), undergraduate (14.8%) or PhD students (10.5%; X^2^ = 16.83, *p* = 0.032). A greater proportion of international students (29.4%) had accessed support compared with 12.2% of domestic students (X^2^ = 32.15, *p* < 0.001). Overall, 20.3% reported accessing support specifically for living costs, such as a COVID hardship grant or grocery vouchers, with no significant differences across degree type (X^2^ = 5.12, *p* = 0 > 0.05) or international student status (X^2^ = 0.94, *p* > 0.05).

The students were invited to tell us more about any burden that they were experiencing as a result of the pandemic. Common responses were: hard adjusting to changed living conditions such as moving back in with parents or living arrangements not conducive to study (e.g., share house situations, whole families working from home); financial pressures including still paying for student accommodation—while not living at the premises—to keep their place; mental health deterioration including anxiety and loneliness; limited time to study due to childcare responsibilities; feeling unmotivated by not having the social aspect of attending on campus; uncertainty about future impacts on academic progression, grades, graduation, and job security; and dissatisfaction with online learning.

Employment was reported as one of the main ways students financially supported themselves while they were studying, with 58.1% having employment throughout the semester and 35.5% throughout the semester break. Around half (48.5%) of students’ employment had been impacted by COVID-19, including reduced hours (16.5%), being stood down—so were not being paid but still had their job (15.1%), or had lost their job (10%). A greater proportion of international students (49.5%) reported the money at their disposal to be less sufficient or not sufficient, compared with 30.2% of domestic students (X^2^ = 23.66, *p* < 0.001).

### 3.3. Psychological Variables

Overall, 34.7% of students reported a sufficient level of wellbeing, while 33.8% reported low wellbeing and another 31.5% very low wellbeing. Wellbeing was highest in postgraduate students (M = 45.75; SD = 23.19), and lowest in undergraduate students (M = 39.19; SD = 19.96; F = 10.67, *p* ≤ 0.001).

Sense of coherence (i.e., feelings about current living situation) was high across all students (M = 3.94; SD = 0.83) and was significantly greater among postgraduate students (M = 4.15; SD 0.96) than undergraduate students (M = 3.90; SD = 0.78, *p* < 0.001). Future anxiety was low across the sample (M = 3.57; SD = 1.17) but was significantly greater among undergraduate compared to postgraduate students (M = 3.66 undergraduate vs. 3.44 postgraduate, *p* = 0.012). Sense of coherence (M = 3.96 international vs. 4.01 home, *p* > 0.05) and wellbeing (M = 41.71 international vs. 41.50 home, *p* > 0.05) were not significantly different between international and domestic students, but future anxiety was higher in international students (M = 3.83 international vs. 3.50 home, *p* = 0.001).

### 3.4. Factors Associated with Low Wellbeing

Table 4 shows three multivariate regression models for sociodemographic factors and low wellbeing. In model 1, we found significant associations between low wellbeing and student age (21–23, OR = 1.84, 95%CI: 1.09–3.11; <20, OR = 1.75, 95%CI: 1.03–2.98), being female (OR = 1.83, 95%CI: 1.30–2.57), low SSS (medium, OR = 1.54, 95%CI: 1.07–2.22; low, OR = 3.56, 95%CI: 2.13–5.95) and speaking a language other than English at home (OR = 1.74, 95%CI: 1.14–2.66). When sense of coherence was added into model 2, all these variables remained associated with lower wellbeing. Students with low sense of coherence showed higher likelihood of low wellbeing (OR = 1.46, 95%CI: 1.06–2.01). Further, when adding future anxiety into model 3, the association lessened between being female (OR = 1.71, 95%CI: 1.19–2.44), having low SSS (OR = 2.79, 95%CI: 1.63–4.77) and student age (21–23, OR = 1.80, 95%CI: 1.04–3.11), but these remained significant. Students with high future anxiety showed higher likelihood of having lower wellbeing (OR = 3.62, 95%CI: 2.58–5.09).

Table 5 shows multivariate regression models for learning experience and low wellbeing. In model 1, we found significant associations between low wellbeing and overall learning experience (negative, OR = 2.94, 95%CI: 2.08–4.17) and impact on study (strongly agree, OR = 3.95, 95%CI: 1.84–8.46). When overall learning experience and impact on study were added to the same model, both remained associated with lower wellbeing.

## 4. Discussion

This study reports the wellbeing of university students in Australia at a time when there were substantial disruptions to the higher education sector due to COVID-19. The required shift to online learning has had a significant (mainly negative) impact on both the overall learning experience of our university students, and their psychological wellbeing. Undergraduate students and those studying for a graduate certificate or diploma, showed greater future anxiety, and lower mental wellbeing and overall life orientation (sense of coherence) than postgraduate students. Future anxiety was also markedly greater in international than domestic students, with international students reporting having accessed COVID-19-related support from the university more than domestic students. Significant predictors of lower wellbeing included being female, having lower subjective social status, lower sense of coherence and higher anxiety, reporting a negative overall learning experience or COVID-19 having a huge impact on their study.

Although we cannot infer that the pandemic influenced differences in wellbeing or whether the association of low wellbeing with other variables could have been detected pre-COVID-19, these findings identify key inequities across students in relation to their learning experience and wellbeing during COVID-19. They highlight the need for universities and policymakers to focus educational, emotional, and financial support to these groups now to avoid exacerbating existing disparities.

The German arm of this collaboration similarly found that future anxiety and sense of coherence was unequally distributed among students with different social backgrounds (Dadaczynski et al.; findings not yet published). They identified gender and subjective social status differences in mental health, with female students and students with lower subjective social status more affected by low wellbeing. This demonstrates that, although the pandemic has been handled differently worldwide, and COVID-19 prevalence is vastly different across countries, the experience of university students may be comparable. Gender differences in COVID-19 related worries have been shown, with female university students scoring significantly higher than male students for depression, anxiety and stress during the early stages of the pandemic [28]. This study also found that younger adult students (aged 18–24 years) had more symptoms of anxiety and depression during COVID-19 than older adult students (≥25 years), supporting our findings of differences between undergraduate and postgraduate students. Although evidence indicates gender and social status differences in stress and anxiety pre-pandemic [29,30], it is possible that the pandemic may deepen these discrepancies. For example, the competing demands of caring responsibilities and online study are more likely to affect women than men, and those with lower social status may not have the same access to the resources they need or adequate internet connectivity to enable them to conduct their studies online. Undergraduate students missed the important socialisation opportunity of orientation week to create social connections which may have contributed to higher prevalence of future anxiety and lower sense of coherence in these students.

Predictors of low wellbeing included those reporting a negative overall learning experience or that COVID-19 had a huge impact on their study. Recent results published from surveys of almost 120 higher education providers in Australia showed that up to 50% of students disliked online learning [31]. Due to the differences in question wording, this cannot be directly compared with our findings, but almost 75% of our sample reported finding it more difficult to learn online, particularly relating to teacher and student interaction. If higher education providers plan to continue with online learning, our findings, together with the report from the Australian government [31], highlight key areas which should be addressed to improve the online learning experiences of students and consequently their wellbeing.

Employment is one of the main ways that students financially supported themselves during their studies. Almost half of the students employed in this sample had their employment impacted by COVID-19. International students in particular invest significant financial sacrifice to undertake their degree, paying significantly higher fees for their tertiary studies than domestic students while, at the same time, being unable to work more than 20 h per week due to visa restrictions [18]. This supports findings from a nationwide survey of 5000 temporary visa holders which found that 60% of international students had lost their job due to the COVID-19 pandemic, and 21% had their hours significantly reduced. In total, 26% were sharing a bedroom to reduce costs, and 46% were financially forced to skip meals on a regular basis [32]. International students may be particularly vulnerable: a recent survey found that they are the most exploited workers with half being paid less than the minimum wage [33]. There is potential that COVID-19 may make this worse due to both students being desperate for income and employers looking to cut costs [34].

Financial support, such as university loans or the temporary suspension of student loan payments could help ease the burden students are currently facing [35], particularly those whose employment has been disrupted during COVID-19. Emotional support is vital for student wellbeing, sense of coherence, and continued learning. Only around 20% of students were making use of support services. Ensuring students are aware of the support services available at universities and beyond is essential and should be featured prominently on University webpages [36]. Practically, students could require support making new living arrangements, where possible. This may be especially important for international students [37], who may have no choice but to return to their home country. Student disengagement can have significant downstream impacts on the research community, slowing essential advances in innovation and knowledge that governments and other organisations rely on.

This large nationwide study investigated the learning experience and wellbeing of university students in Australia during the COVID-19 pandemic. The cross-sectional design of the study and the self-selective (convenience) sample mean that we cannot draw causal conclusions or express that these findings generalise to the whole student body in Australia, but it does provide us with some indication of which students may be most vulnerable. Compared to the whole population of Australian university students, our sample had a higher representation of female (67% vs. 55.6%) and postgraduate students (36.9% vs. 30.2%), but fewer international students (22.9% vs. 32.4%) [38]. Some sociodemographic groups—potentially those more likely to perceive distress—may not be accounted for in this sample. In addition, regional variation in university COVID-19 policies across Australia may have influenced student experiences of the pandemic, however examining this was beyond the scope of our research. The sample had a large proportion of students studying medical and health sciences which may impact our findings as medical students typically have higher anxiety [39] and may be those students who have experienced the greatest disruptions from online learning. Results for the overall sense of coherence scale were reported rather than subscales, as Cronbach’s Alpha was low for the manageability and comprehensibility subscales. The scale, therefore, may not be consistently measuring sense of coherence and the findings should be interpreted with caution.

## 5. Conclusions

Supporting the health, wellbeing, and learning experience of all students should be of highest priority now and post-pandemic. Strategies specifically targeting female students and those with low self-reported social status are urgently needed to avoid exacerbating existing disparities. This could be accomplished by strengthening individual resources and capacities, and creating environments which are supportive, responsive and needs-orientated.

## Figures and Tables

**Table 1 ijerph-18-00866-t001:** Detailed description of the measures used.

Measure	Description of Measure	Response Options/Range and Interpretation	Reliability Analysis (Cronbach’s Alpha)
Wellbeing [23]	Self-perceived wellbeing for past two weeks (e.g., over the last 2 weeks I have felt active and vigorous)	Six-point response scale (0 = at no time, 1 = some of the time, 2 = less than half of the time, 3 = more than half of the time, 4 = most of the time, 5 = all of the time) Raw score multiplied by 4 = 0 lowest wellbeing—to 100 highest Further scored as ≤28 very low wellbeing, ≤50 low wellbeing, >50 high wellbeing	0.890
Sense of Coherence [24]	How do you personally find your current living situation in general?	Nine-items with three subscales: a behavioural component (manageability) (2 items), a motivational component (meaningfulness) (3 items), and a cognitive component (comprehensibility) (4 items). Scored 1 to 7 with higher values indicating higher sense of coherence	Overall = 0.538
Future Anxiety [25]	Nine items: First five items from short version of future anxiety scale “Dark Future Scale” and four items from the long versionExample: I am afraid that the problems which trouble me now will continue for a long time.	Seven-point response scale (0 = decidedly false, 1 = false, 2 = somewhat false, 3 = hard to say, 4 = somewhat false, 5 = true, 6 = decidedly true) Higher scores indicate higher anxiety	Dark future scale = 0.838 Extended dark future scale = 0.817
Subjective Social Status (SSS) [22]	Where would you place yourself on this ladder? Please mark a field from 1–10 where you think you stand at this time in your life relative to other people in Australia	Ten-point scale (1 to 10) with higher points indicating a higher subjective social status Further scored as low SSS (1–4), medium SSS (5–7), high SSS (8–10)	

**Table 2 ijerph-18-00866-t002:** Sample characteristics (N = 787).

Sample	N (%)
**Age: mean (SD)**	25.74 (9.24)
**Gender**	
Male	251 (31.9)
Female	527 (67.0)
Other/prefer not to say	9 (1.1)
**State/Territory**	
Australian Capital Territory	45 (5.7)
Northern Territory	6 (0.8)
New South Wales	449 (57.1)
Victoria	110 (14.0)
Queensland	75 (9.5)
Western Australia	49 (6.2)
South Australia	28 (3.6)
Tasmania	25 (3.2)
**Born in Australia: yes**	514 (65.3)
**International student: yes**	180 (22.9)
**Main language at home: English**	628 (79.8)
**Aboriginal or Torres Strait Islander: yes**	10 (1.2)
**Socioeconomic position (1–10)**	6.07 (1.87)
**Top 4 universities represented**	
The University of Sydney	242 (30.7)
University of New South Wales	47 (6.0)
Australian National University	40 (5.1)
University of Queensland	30 (3.8)
**Top 4 areas of study**	
Medicine/Health Sciences	323 (41.0)
Business	64 (8.1)
Arts and Social Sciences	61 (7.8)
Maths/Natural Sciences	61 (7.8)
**Degree type**	
Undergraduate Bachelor	461 (58.6)
Postgraduate—Masters	166 (21.1)
Postgraduate—PhD	124 (15.8)
Graduate Certificate/Diploma	23 (2.9)
Other	13 (1.7)

**Table 3 ijerph-18-00866-t003:** Impact of COVID-19 on living and study (N = 787).

	N (%)
**Impact on living: yes**	181 (23.0)
Returned to home country	16 (2.0)
Unable to travel to Australia	7 (0.9)
Moved out of student accommodation	77 (9.8)
Other	97 (12.3)
**Accessed support offered by university: yes**	127 (16.1)
**COVID-19 has had huge impact on my study last 2 weeks**	
Agree *	683 (86.8)
Neither agree nor disagree	26 (3.3)
Disagree	78 (9.9)
**Accessed extra support for living costs: yes**	160 (20.3)
**How impacted**	
Units shifted online	524 (66.6)
Units of study cancelled	49 (6.2)
Units of study postponed	73 (9.3)
Changes to assessment format	460 (58.4)
Changes to assessment timelines	398 (50.6)
Changes to frequency of contact with teachers	436 (55.4)
Other	180 (22.9)
**Impact on overall learning experience**	
Positive	50 (6.4)
Negative	558 (70.9)
Neutral	179 (22.7)
**I find it more difficult to learn online than face-to-face: agree ***	588 (74.7)
**I prefer online learning to face-to-face: agree**	217 (27.6)
**I find it hard to interact with teachers online: agree**	587 (74.6)
**I find it hard to interact with other students online: agree**	666 (84.6)
**My internet is unreliable and disrupts online learning: agree**	410 (52.1)
**I find online learning less time consuming than face to face: agree**	375 (47.6)
**My home environment supports online learning: agree**	435 (55.3)
**I am confident with my computer skills: agree**	672 (85.4)

* agree includes strongly agree, agree, agree a bit.

**Table 4 ijerph-18-00866-t004:** Multivariable model examining sociodemographic factors associated with lower (<50) wellbeing (N = 765).

	Adjusted Odds Ratios (OR)					
	Model 1		Model 2		Model 3	
	OR	95%CI	OR	95%CI	OR	95%CI
**Age**						
≥27 years	(ref.)		(ref.)		(ref.)	
24–26 years	1.199	0.717–2.005	1.143	0.681–1.917	0.931	0.539–1.608
21–23 years	1.844	1.092–3.114	1.816	1.074–3.072	1.799	1.039–3.112
≤20 years	1.750	1.027–2.982	1.721	1.008–2.938	1.683	0.964–2.937
**Gender**						
Male	(ref.)		(ref.)		(ref.)	
Female	1.828	1.302–2.568	1.887	1.339–2.658	1.705	1.193–2.437
**Subjective Social Status**						
High (8–10)	(ref.)		(ref.)		(ref.)	
Medium (5–7)	1.538	1.066–2.219	1.492	1.032–2.158	1.410	0.959–2.074
Low (1–4)	3.560	2.130–5.948	3.518	2.100–5.892	2.789	1.632–4.767
**Language spoken at home**						
English	(ref.)		(ref.)		(ref.)	
Other	1.741	1.140–2.658	1.744	1.142–2.664	1.422	0.909–2.225
**Degree type**						
Postgraduate	(ref.)		(ref.)		(ref.)	
Undergraduate/graduate certificate/diploma	1.202	0.750–1.925	1.177	0.734–1.888	1.066	0.650–1.747
**Sense of Coherence**						
High			(ref.)		(ref.)	
Low			1.459	1.057–2.013	1.510	1.079–2.111
**Future Anxiety**						
Low					(ref.)	
High					3.623	2.578–5.092

**Table 5 ijerph-18-00866-t005:** Multivariable model examining learning experience associated with lower (<50) wellbeing (N = 765).

	Adjusted Odds Ratios (OR)			
	Model 1		Model 2	
	OR	95%CI	OR	95%CI
**Impact on living**				
Yes	(ref.)			
No	0.751	0.525–1.076		
**Accessed support**				
No	(ref.)			
Yes	1.511	0.990–2.305		
Not aware of any support	1.546	0.974–2.453		
**Overall learning experience**				
Neutral	(ref.)		(ref.)	
Positive	0.721	0.381–1.363	0.670	0.346–1.299
Negative	2.941	2.077–4.165	2.101	1.433–3.078
**Impact on study**				
Strongly disagree	(ref.)		(ref.)	
Disagree/Disagree a bit	0.471	0.182–1.126	0.438	0.166–1.158
Neither agree nor disagree	1.829	0.629–5.316	1.736	0.584–5.164
Agree a bit/Agree	1.780	0.842–3.760	1.236	0.566–2.696
Strongly agree	3.945	1.839–8.462	2.301	1.025–5.168
**Accessed living costs support**				
Yes	(ref.)			
No	0.854	0.589–1.237		

## Data Availability

The data presented in this study are available on request from the corresponding author. The data are not publicly available because participants did not consent to having their data made publicly available.

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
