# Peer review of "Psychological Wellbeing and Academic Experience of University Students in Australia during COVID-19"

_ijerph, 2021, doi:10.3390/ijerph18030866_

Round 1

Reviewer 1 Report

COVID-19 pandemic has become the major event of 2020 that has affected and continues to exert unprecedented influence on all aspects of the activity of both society and individuals in the whole world.

In order to successfully overcome the consequences of the pandemic, efforts of both the authorities and researchers should be aimed at maintaining the well-being of the most vulnerable groups of the population.

A fair number of studies already conducted since the start of the COVID-19 pandemic show that one of these groups, along with the elderly, is young people. In this regard, there is a comment to the authors of the article - in the introduction, a broader review of the latest data on the study area would be desirable.

This research focuses on young people - students who are at the stage of self-determination, entering an independent life, and the pandemic has certainly increased the state of uncertainty, including that associated with their future. The analysis revealed a low level of wellbeing among most students during a pandemic; differences were found between undergraduate and postgraduate students.

Although the manuscript is recommended for publication, I would like to make a few comments:

  1. The authors note that "although evidence indicates gender and social differences in stress and anxiety pre-pandemic, the impacts of the pandemic may deepen these discrepancies”. However, this is an assumption, and there is no data on student characteristics before the pandemic in this study. This raises the question of whether it was the factor of the pandemic that influenced wellbeing, or whether the association of low wellbeing with gender, low subjective social status, and negative overall learning experiences could have been detected before the pandemic.

        Besides, maybe there is a common factor behind these characteristics?

  1. The same questions arise when discussing comparisons between undergraduate and postgraduate students. Which differences in characteristics are caused by the pandemic and which exist between these groups of students regardless of the pandemic situation?
  2. The authors themselves point out the limitations of the sample. Almost half of the students in the sample study medical and health sciences. The authors point out that this may influence findings as medical students typically have higher anxiety. But it is these areas of study that have restrictions in online learning, and students may experience missed learning opportunities, which may distort the results obtained.
  3. In many countries, in different regions, there was a different situation related to the pandemic. I do not know the details of the situation in Australia, but if it varied by region, then this could be additional factors of wellbeing.

Since this study is part of a large-scale international survey, it seems that the analysis of all the data obtained will broaden the discussion and allow a more accurate interpretation of the results.

In any case, the practical relevance of this study is obvious, and the results can make an important contribution to solving the current problems of the COVID-19 pandemic.

Author Response

COVID-19 pandemic has become the major event of 2020 that has affected and continues to exert unprecedented influence on all aspects of the activity of both society and individuals in the whole world.

In order to successfully overcome the consequences of the pandemic, efforts of both the authorities and researchers should be aimed at maintaining the well-being of the most vulnerable groups of the population.

A fair number of studies already conducted since the start of the COVID-19 pandemic show that one of these groups, along with the elderly, is young people. In this regard, there is a comment to the authors of the article - in the introduction, a broader review of the latest data on the study area would be desirable.

Author response: We have now expanded the introduction to include more specific research about psychological well-being in students and believe we have presented some of the latest data (lines 63-79) on the study area looking at stress for students during COVID-19. We do not want to expand to research in younger people more generally, as the objectives of this research were to examine the experiences of university students specifically due to the much-changed nature of their studies.

This research focuses on young people - students who are at the stage of self-determination, entering an independent life, and the pandemic has certainly increased the state of uncertainty, including that associated with their future. The analysis revealed a low level of wellbeing among most students during a pandemic; differences were found between undergraduate and postgraduate students.

Although the manuscript is recommended for publication, I would like to make a few comments:

  1. The authors note that "although evidence indicates gender and social differences in stress and anxiety pre-pandemic, the impacts of the pandemic may deepen these discrepancies”. However, this is an assumption, and there is no data on student characteristics before the pandemic in this study. This raises the question of whether it was the factor of the pandemic that influenced wellbeing, or whether the association of low wellbeing with gender, low subjective social status, and negative overall learning experiences could have been detected before the pandemic.

Author response: Thank you for highlighting this important point. We have softened this sentence to ‘although evidence indicates gender and social differences in stress and anxiety pre-pandemic, it is possible that the pandemic may deepen these discrepancies’ and we then go on to give some examples as to why this might be the case. We have also acknowledged at the end of the paragraph that we do not have any data on these students pre-pandemic and can therefore not infer causality.  

Lines 298-301: ‘We did not however, collect data from the student participants pre-pandemic and can therefore not infer that the pandemic influenced wellbeing, or whether the association of low wellbeing with other variables could have been detected before the pandemic.’  

        Besides, maybe there is a common factor behind these characteristics?

  1. The same questions arise when discussing comparisons between undergraduate and postgraduate students. Which differences in characteristics are caused by the pandemic and which exist between these groups of students regardless of the pandemic situation?

Author response: We have now acknowledged this in response to your point above.

  1. The authors themselves point out the limitations of the sample. Almost half of the students in the sample study medical and health sciences. The authors point out that this may influence findings as medical students typically have higher anxiety. But it is these areas of study that have restrictions in online learning, and students may experience missed learning opportunities, which may distort the results obtained.

Author response: Thank you for this important point. We have now included this in our limitations (lines 355-356). ‘The sample had a large proportion of students studying medical and health sciences which may impact our findings as medical students typically have higher anxiety [33] and may be those students who have experienced the greatest disruptions from online learning.’

  1. In many countries, in different regions, there was a different situation related to the pandemic. I do not know the details of the situation in Australia, but if it varied by region, then this could be additional factors of wellbeing.

Author response: Yes, the situation across Australia did differ depending on which state or territory you lived in, with the restrictions easing earlier in Western Australia and the Northern Territory than other states. However, the number of participants from those two states (n=55) was small and therefore unlikely to have much impact on overall wellbeing. Also, at the time of the survey, the majority of students were still studying online as universities were still closed and therefore we do not believe this will have been a strong factor involved in wellbeing. We have added a sentence to our limitations to reflect this (line 353-355): In addition, regional variation in university COVID-19 policies across Australia may have influenced student experiences of the pandemic, but examining this was beyond the scope of our research.

Since this study is part of a large-scale international survey, it seems that the analysis of all the data obtained will broaden the discussion and allow a more accurate interpretation of the results.

In any case, the practical relevance of this study is obvious, and the results can make an important contribution to solving the current problems of the COVID-19 pandemic.

Reviewer 2 Report

The article has a wide scope with reference to the university population in the situation dominated by the COVID-19 effect. It proposes a descriptive analysis consistent with the proposals defined by the research questions.
It should be noted that the weaknesses of the study are also described by the authors, specifically those related to the sample and to the instrument for obtaining information.
It would have been interesting to relate the well-being analyzed with other dimensions of quality of life, which would make it possible to explain the situation in which university students find themselves in a more holistic way.

On the article under review, I state the following:

  1. The article is based on an international study, specifically it is contextualized in Australia. The opportunity to propose a study in the framework of COVID-19 on the "Psychological well-being and academic experience of university students in Australia during COVID-19" is understood. In relation to this, in writing, it would also be appropriate to support this need (what?) and its functionality (why?).
  2. The intended objectives and their justification are not clearly established, so as to make it easier to deduce the choice of the methodological approach and the subsequent delimitation of results. In this regard, the intended objectives are not described, choosing to pose only questions. Assessing the impact of COVID-19 requires a methodological approach that is consistent with the objectives of this assessment, and justifies them.
  3. The conclusions are set on the attention that should be given to students in situations that generate vulnerability; However, they should also be specified based on the issues specified, or better on the objectives that have not been exposed in this writing, even if they are inferred.

Strengths:

- The topic is current and necessary to guide on support actions by higher education educational institutions.

- Looking at the explicit issues, the study focuses on them.

- The results are clearly described, they are associated with the research questions posed. The relevant procedures are used to achieve them.

Weaknesses:

- Background could be expanded to explain the research questions. The theoretical development proposed in the introduction is limited to what is related to COVID-19, as well as to describe some references in relation to well-being, without affecting its conceptualization and its establishment parameters.

- The sample, as expressed in the writing on the limitations, being for convenience, obviously raises problems of generalization of the results.

- Contextualization, given that it refers to 4 prestigious universities, can only be circumscribed to them, so it is possible to ask about the incidence in other universities that lack the resources or have other policies on quality standards other than those used in these universities. It should fall within the framework of the limitations.

- Regarding the instrument, the analysis of the reliability (Cronbach's alpha) of the factor "sense of coherence" is excessively low, so some decision should have been taken in this regard to strengthen the value of the factor. In the brief it is considered within the framework of the limitations of the investigation.

Author Response

The article has a wide scope with reference to the university population in the situation dominated by the COVID-19 effect. It proposes a descriptive analysis consistent with the proposals defined by the research questions.
It should be noted that the weaknesses of the study are also described by the authors, specifically those related to the sample and to the instrument for obtaining information.
It would have been interesting to relate the well-being analyzed with other dimensions of quality of life, which would make it possible to explain the situation in which university students find themselves in a more holistic way.

Author response: Thank you for suggesting potential further avenues for research, looking at other dimensions of quality of life. Due to the international collaboration, the measures used were predetermined and included to allow for international comparisons.

On the article under review, I state the following:

  1. The article is based on an international study, specifically it is contextualized in Australia. The opportunity to propose a study in the framework of COVID-19 on the "Psychological well-being and academic experience of university students in Australia during COVID-19" is understood. In relation to this, in writing, it would also be appropriate to support this need (what?) and its functionality (why?).

Author response: We are unsure what the reviewer intends to be changed based on this comment. Our objective was to identify student groups most impacted by the pandemic to enable targeted support and systems in place moving forward to minimise impact on student learning and wellbeing. This has been clarified on lines 82-86 of the introduction.

  1. The intended objectives and their justification are not clearly established, so as to make it easier to deduce the choice of the methodological approach and the subsequent delimitation of results. In this regard, the intended objectives are not described, choosing to pose only questions. Assessing the impact of COVID-19 requires a methodological approach that is consistent with the objectives of this assessment, and justifies them.

Author response: We have now updated the introduction, as well as describing our intended objectives and research questions.

  1. The conclusions are set on the attention that should be given to students in situations that generate vulnerability; However, they should also be specified based on the issues specified, or better on the objectives that have not been exposed in this writing, even if they are inferred.

Author response: We are unsure what the reviewer intends to be changed based on this comment. Our objective was to identify student groups most impacted by the pandemic to enable targeted support and systems in place moving forward to minimise impact on student learning and wellbeing. This has been clarified on lines 82-86 of the introduction.

Strengths:

- The topic is current and necessary to guide on support actions by higher education educational institutions.

- Looking at the explicit issues, the study focuses on them.

- The results are clearly described, they are associated with the research questions posed. The relevant procedures are used to achieve them.

Author response: Thank you for highlighting the strengths of our study.

Weaknesses:

- Background could be expanded to explain the research questions. The theoretical development proposed in the introduction is limited to what is related to COVID-19, as well as to describe some references in relation to well-being, without affecting its conceptualization and its establishment parameters.

Author response: We have now updated the introduction and given more references in relation to wellbeing in students, as well as describing our intended objectives and research questions.

- The sample, as expressed in the writing on the limitations, being for convenience, obviously raises problems of generalization of the results.

Author response: Yes, this has been acknowledged in the limitations.

- Contextualization, given that it refers to 4 prestigious universities, can only be circumscribed to them, so it is possible to ask about the incidence in other universities that lack the resources or have other policies on quality standards other than those used in these universities. It should fall within the framework of the limitations.

Author response: The 4 universities referred to in Table 2 are the top 4 universities represented. In total we had students respond from 41 of the 43 universities across Australia. We have acknowledged on lines 353-355 that there may be some regional variation that may have influenced student experiences of the pandemic: In addition, regional variation in university COVID-19 policies across Australia may have influenced student experiences of the pandemic, but this was beyond the scope of our research.

- Regarding the instrument, the analysis of the reliability (Cronbach's alpha) of the factor "sense of coherence" is excessively low, so some decision should have been taken in this regard to strengthen the value of the factor. In the brief it is considered within the framework of the limitations of the investigation.

Author response: This was also commented on by another reviewer and after discussion with the broader team, we have decided to use only the overall scale for sense of coherence due to the low reliability of some of the subscales and have mentioned this in our limitations (line 380-382).

We reported results for the overall sense of coherence scale rather than subscales, as Cronbach’s Alpha was low for the manageability and comprehensibility subscales.

Reviewer 3 Report

Abstract

  • Please should fit APA style. It is not usual to insert direct results in this section

Introduction

  • It is suggested to establish sections in the review to expose the previous evidence of the relationships between variables.
  • The rationale is brief and inconsistent. Although different data on the impact of the pandemic are exposed, the constructs and models to be tested are not clearly established. For example,

1) Psychological well-being is a well-defined construct based on previous research. It is also not evaluated on the instruments as such.

2) The teaching-learning processes at the University are not substantiated or evaluated.

  • The hypotheses must be rearranged. It seems reasonable that those referring to socio-demographic variables are prior to the others.

Method

  • This work has numerous methodological problems, unsolved. It should be restructured in the ordered and necessary sections

1) Participants: selection and exclusion criteria.

2) Instruments: they have measurement problems with low reliability and validity without reporting

3) Procedure: The permission of the Ethics Committee should be referred to here.

4) Data analysis: design and analysis carried out according to hypotheses

Results

  • The characteristics of the sample should be in the "Participants" section.
  • The results are interesting, but excessively descriptive and very basic.

Discussion

  • The discussion and conclusions that can be reached with this type of design are very basic
  • Possible inferences are limited.

References

They should be updated and complet

Author Response

Introduction
It is suggested to establish sections in the review to expose the previous evidence of the relationships between variables.

Author response: Thank you for this suggestion. We would prefer however not to include subheadings in the introduction to allow this to flow easily.  We have added a paragraph (lines 55-62, and below) including evidence on psychological wellbeing to enhance the link between the variables.

The rationale is brief and inconsistent. Although different data on the impact of the pandemic are exposed, the constructs and models to be tested are not clearly established. For example,
1) Psychological well-being is a well-defined construct based on previous research. It is also not evaluated on the instruments as such.

Author response: We have now expanded the introduction to include more specific research about psychological well-being in students (lines 55-62). As noted in the Methods, we used the World Health Organisation wellbeing index to measure psychological wellbeing.

University students have been identified as a ‘very high risk population’ for mental health difficulties [8]. Large studies conducted in Australian universities have reported elevated levels of generalized psychological distress and severe depressive symptoms when compared to population samples [9,10]. Studies in the United States and United Kingdom have reported similar findings among tertiary students [11, 12]. Psychological distress negatively impacts on student learning, participation, and their experience of university life, so it is important for universities to understand the student experience of particular stressors to better support their psychological wellbeing [13].

2) The teaching-learning processes at the University are not substantiated or evaluated.

Author response: We are unsure what the reviewer intends to be changed based on this comment. We have expanded to specify that the university transitioned rapidly from face-to-face, to online delivery (line 49). We then evaluated the (self-reported) impact of these changes on our study participants.

The hypotheses must be rearranged. It seems reasonable that those referring to socio-demographic variables are prior to the others.

Author response: The current order of the research questions reflects the order of analysis, as research question 1 is describing the sample and therefore provides background for research questions 2 and 3. We would therefore prefer to retain the current order, but will make the adjustments if required by the editorial team.

Method
This work has numerous methodological problems, unsolved. It should be restructured in the ordered and necessary sections
1) Participants: selection and exclusion criteria.

Author response: This has now been changed.

2) Instruments: they have measurement problems with low reliability and validity without reporting.

Author response: We have now changed the section heading to ‘instruments’ and have decided to use only the overall scale for sense of coherence due to the low reliability of some of the subscales and have mentioned this in our limitations (line 380-382).

We reported results for the overall sense of coherence scale rather than subscales, as Cronbach’s Alpha was low for the manageability and comprehensibility subscales.

3) Procedure: The permission of the Ethics Committee should be referred to here.

Author response: This has now been changed.

4) Data analysis: design and analysis carried out according to hypotheses.

Author response: We have now referred specifically in this section to which analysis corresponded with which research question.

We have also restructured the methods section to the order suggested above.

Results
The characteristics of the sample should be in the "Participants" section.

Author response: We have now included a ‘participants’ subheading.

The results are interesting, but excessively descriptive and very basic.

Author response: We are unsure what the reviewer intends to be changed based on this comment.

Discussion
The discussion and conclusions that can be reached with this type of design are very basic

Author response: We are unsure what the reviewer intends to be changed based on this comment.

We have noted in our limitations that we cannot draw causal conclusions or express that these findings generalise to all university students due to the design of this study.

Possible inferences are limited.

Author response: We are unsure what the reviewer intends to be changed based on this comment.

References
They should be updated and complete.

Author response: We have included some additional references and updated the latest Australian COVID-19 data.

Reviewer 4 Report

A fine work, to which I have few remarks, mainly concerning the Limitations section.

The main comparison was with German data from the same project. Interestingly, in Introduction you mentioned a Chinese study with similar findings as yours. While the pandemic has similar effects on the stress caused to students in both capitalist and socialist countries, I believe the latter are better equiped to face it due to the buffer offered by the economic system. You might wish to comment on this. Furthermore, the epidemic is to s certain extent under control in Australia, but out of control in Germany.

Moreover, the participation of Aborigens was low. Could this have been due to a generally lower educational level of this Australian minority or to limited access to online surveying? Despite your claim of representativeness of your sample, there might have been factors limiting the access to the survey of more disadvantaged people with a higher likelihood to perceived distress. You should comment on this.

By the way, deaths in Australia are already 909. This is to stress the fleeting narìture of your data, that needs to be underlined.

Author Response

A fine work, to which I have few remarks, mainly concerning the Limitations section.

The main comparison was with German data from the same project. Interestingly, in Introduction you mentioned a Chinese study with similar findings as yours. While the pandemic has similar effects on the stress caused to students in both capitalist and socialist countries, I believe the latter are better equiped to face it due to the buffer offered by the economic system. You might wish to comment on this. Furthermore, the epidemic is to s certain extent under control in Australia, but out of control in Germany.

Author response: Yes, we found it interesting that the German data showed similar findings given the difference in the impact of the pandemic with Australia. We have added a comment to reflect this in the discussion (line 281-284). ‘This demonstrates that although the pandemic has been handled differently worldwide, and COVID-19 prevalence is vastly different across countries, the experience of university students may be comparable.’

Although you raise an interesting point about the difference between capitalist and socialist countries, we feel this is beyond the scope of the objectives of this research.

Moreover, the participation of Aborigens was low. Could this have been due to a generally lower educational level of this Australian minority or to limited access to online surveying? Despite your claim of representativeness of your sample, there might have been factors limiting the access to the survey of more disadvantaged people with a higher likelihood to perceived distress. You should comment on this.

Author response: The percentage of ABTSI students in our sample (1.2%) is actually comparable to the number in higher education in 2017 (1.8%, https://www.aihw.gov.au/reports/australias-welfare/higher-education-and-vocational-education). We have now acknowledged in our limitations that other factors may have limited access to the survey in other groups (lines 350-353): Due to the potential that not all socioeconomic groups would have had the same level of access to the survey due to its online nature, some groups more likely to perceive distress may not be accounted for in the sample. 

By the way, deaths in Australia are already 909. This is to stress the fleeting narìture of your data, that needs to be underlined.

Author response: We are unsure what the reviewer means to state by this comment. We have updated the statistics so that they are as up-to-date as possible.

Round 2

Reviewer 3 Report

The manuscript has improved substantially.